# Novel Shared Heritable Candidate Risk Loci of Breast and Endometrial Cancer—A Swedish Haplotype Genome-Wide Association Study

**DOI:** 10.3390/ijms26125461

**Published:** 2025-06-06

**Authors:** Elin Barnekow, Wen Liu, Mikael Andersson Franko, Anna von Wachenfeldt, Camilla Wendt, Emma Tham, Miriam Mints, Tracy A. O’Mara, Per Hall, Sara Margolin, Annika Lindblom

**Affiliations:** 1Department of Clinical Science and Education, Karolinska Institutet, 11883 Stockholm, Swedenanna.vonwachenfeldt-vappling@regionstockholm.se (A.v.W.);; 2Department of Oncology, Södersjukhuset, 11883 Stockholm, Sweden; 3Department of Molecular Medicine and Surgery, Karolinska Institutet, 17176 Stockholm, Sweden; 4Department of Neuroscience, Uppsala University, 75237 Uppsala, Sweden; 5Department of Clinical Genetics, Karolinska University Hospital, 17164 Stockholm, Sweden; 6Department of Women’s and Children’s Health, Karolinska University Hospital, 17177 Stockholm, Sweden; 7Cancer Research Program, QIMR Berghofer, Brisbane, QLD 4006, Australia; tracy.omara@qimrberghofer.edu.au; 8Department of Medical Epidemiology and Biostatistics, Karolinska Institutet, 17165 Stockholm, Sweden

**Keywords:** breast cancer, endometrial cancer, GWAS, haplotype, genetic risk loci

## Abstract

Breast and endometrial cancer are prevalent and share both hormonal and environmental risk factors. This study aimed to identify shared germline genetic risk loci for these cancers. In total, 1116 endometrial cancer cases, 3200 breast cancer cases, and 5021 healthy controls were included in a merged sliding window haplotype genome-wide association study (GWAS). This analysis employed a logistic regression model in PLINK v1.07. The results from this merged analysis were compared with previous individual analyses of the same samples. The analysis identified three loci that influenced both the risk of breast and endometrial cancer: 8p21.1 (OR 2.1; *p* 1.6 × 10^−8^), 16q24.3 (OR 2.4; *p* 3.8 × 10^−8^) and 17q11.2 (OR 1.3; *p* 4.3 × 10^−8^). This combined haplotype GWAS of endometrial and breast cancers identified three loci associated with shared genetic risk, two of which were novel: 16q24.3 and 17q11.2. Further studies are warranted to replicate these findings and to determine its pathophysiological role and future clinical implications.

## 1. Introduction

Breast (BC) and endometrial cancer (EC) affect over 2.7 million women annually [1]. Both cancers are estrogen-dependent tumors and share hormonal and environmental risk factors. Key risk factors include early menarche, late menopause, delayed first birth, nulliparity, prolonged menopausal hormone therapy, and obesity, all of which contribute to an increased risk of these cancers [2,3,4,5,6,7].

Moderate to high-risk germline pathogenic variants contribute to BC and EC risk independently. In many countries, variant screening of susceptibility genes (*BRCA1*, *BRCA2*, *PALB2*, *TP53*, *STK11*, *CDH1*, *PTEN*, *ATM*, *BARD1*, *RAD51C*, *RAD51D* and *CHEK2* for BC and the DNA mismatch-repair genes, *MLH1*, *MSH2*, *MSH6*, and *PMS2*, in Lynch syndrome for EC) are offered in clinical practice of high-risk women [8,9,10]. Furthermore, genome-wide association studies (GWASs) have identified susceptibility loci that individually confer a low increase in risk for both cancers [11,12].

Von Wachenfeldt et al., 2007 demonstrated an over-representation of EC among Swedish families with a history of BC, which was further supported by a tumor spectrum study of non-BRCA BC families and an epidemiological registry-based study on familial cancer from the Swedish Multi-Generation registry [13,14,15]. A large American cohort study also revealed an overrepresentation of EC in individuals with prior breast cancer diagnosis, even after adjusting for shared environmental risk factors, suggesting a shared genetic risk could be involved [16]. Thus, besides environmental risk factors, EC and BC may share genetic susceptibility loci. Despite known variants in *PTEN*, which are rare (1:200,000), much remains unclear regarding the shared heritable factors between EC and BC, highlighting the need for new approaches to address this question [17].

Previously, we demonstrated the feasibility of using haplotype analysis alongside SNP analysis to identify novel risk loci for colorectal cancer, EC and BC independently within Swedish study populations [18,19,20]. By applying the same method and a comparable sample size, we conducted a haplotype GWAS to identify chromosomal regions that may harbor potential shared susceptibility loci for EC and BC. This analysis included Swedish cases of EC (*N* = 1116) and BC (*N* = 3200), along with controls (*N* = 5021). These samples were previously used together with various populations in large independent SNP GWASs in collaboration with the Endometrial Cancer Association Consortium (ECAC) and the Breast Cancer Association Consortium (BCAC) [11,12]. To confirm a combined effect of EC and BC, we compared the results with previous individual analyses of the same samples [18,19].

## 2. Results

In the combined analysis of EC and BC, we identified three candidate risk loci: 8p21.2, 16q24.3 and 17q11.2. In stage one, we identified six significant susceptibility loci for EC and BC on chromosome 8, 10, 11, 16 (two loci) and 17 (Table 1, Appendix A).

In stage two, we compared the results of identical haplotypes from the present combined EC–BC analysis with the previous individual analysis of the same EC and BC samples and controls to examine a potential combined effect [18,19]. Three of the loci—8p21.2, 16q24.3, and 17q11.2—demonstrated lower *p*-values in the combined analysis compared to the previous individual EC and BC analyses, suggesting a shared genetic risk within these loci (Table 2). Notably, the latter loci did not show significant associations in previous independent analyses. In contrast, three well-known BC risk loci—10q26.13, 11q13.3, and 16q12.1—exhibited higher *p*-values in the combined analysis compared to previous individual BC analyses, indicating that the inclusion of EC samples diluted the association in these three regions. Moreover, one of the loci with combined EC–BC effect spanned multiple genes, and two loci spanned a single gene (Table 1, Figure 1).

## 3. Discussion

We identified three candidate risk loci associated with EC and BC at 8p21.2, 16q24.3 and 17q11.2. The odds ratios (ORs) for these novel loci were similar in both the combined and individual analyses. The larger combined analysis likely allowed these regions to reach the significant threshold, which was not achievable in the individual analyses due to the rarity of the haplotypes, particularly at 8p21.2 and 16q24.3 (Table 1). The locus at 8p21.2 has previously been identified in an individual BC GWAS, using the same BC samples [18]. The other three significant loci (the known BC loci 10q26.13, 11q13.3, and 16q12.1) exhibited higher *p*-values in the combined analysis compared to the previous individual analyses of EC and BC, suggesting no shared EC–BC effect in these loci (Table 2) [11,19]. This was consistent with two targeted EC GWAS investigating a total of 16 known BC susceptibility loci, which reported no significant or inverse association between EC and BC at these loci [21,22]. Replication of the three novel candidate EC–BC loci, along with complementary analyses, such as sequencing and functional studies, is warranted for future research.

Initially, known hereditary cancer syndromes and their associated pathogenic variants were thought to be related to a single tumor type, such as BRCA1/2 in BC and APC or the DNA mismatch repair genes in colorectal cancer [23,24,25,26]. However, it has become increasingly clear that these variants are linked to an elevated risk across multiple tumor types [27,28,29]. To our knowledge, this GWAS is the only study employing a genome-wide syndrome approach that combines EC and BC to search for shared genetic risk. However, a meta-analysis of GWASs across various tumor types with a targeted approach, including EC and BC, identified associations between EC and BC with three SNPs on chromosome 2 and inverse association with one SNP on the same chromosome [30]. Notably, none of these four SNPs were analyzed in this study. Furthermore, EC is the most frequently observed secondary malignancy among BC survivors, which may partly reflect the adjuvant use of tamoxifen, known for its pro-estrogenic effect on the endometrium [31,32,33]. However, Sung et al. demonstrated a similar increase in risk following hormone receptor-negative BC, which is not treated with tamoxifen, suggesting that this increase may represent a shared genetic risk [34].

Several genes located in the three novel EC–BC loci have previously been discussed in the context of cancer, although many have not been reported to be involved in EC and BC. The gene *BNIP3L* at 8p21.2 has been suggested to have a tumor suppressive effect in various cancer types [35]. The locus 8p21.2 was early detected with loss of heterozygosity (LOH) in BC; prostate and ovarian cancer and somatic deletions in this region have been identified in EC and ovarian cancer [36,37,38,39]. Previous individual BC GWASs identified one significant haplotype at 8p21, while the present combined EC–BC GWAS revealed eight with a shared central haplotype and similar ORs but with various lengths, representing the same genetic loci. The mitophagy receptor BNIP3 is highly expressed in various cancers, including BC and EC, and functions as a tumor suppressor gene [40,41]. The gene *BNIP3L* (BNIP3-like) shares similar properties with *BNIP3* [40,42]. One study using HCT116 cells demonstrated that *BNIP3* is activated by the estrogen receptor beta (*ERβ*), promoting autophagy [43]. Furthermore, ERβ-mediated degradation of *CyclinD1* was suggested to inhibit colon cancer cell growth via autophagy [43]. 

The locus at 16q24.3 is a known region of LOH associated with BC and prostate cancer [44,45]. This region contains five genes (Table 1, Figure 1). Among them, *MVD*, a member of the mevalonate pathway, interacts with *RAC3* in HER2-positive breast cancer [46]. When hypomethylated, *RAC3* can promote cell proliferation and invasion by increasing *FASN* expression in EC, potentially serving as a link between these two cancer types. [47]. The genes *SNAI3* and *PIEZO1*, which are known to promote epithelial–mesenchymal transition (EMT), are noteworthy. EMT is a process that enables epithelial cells to acquire migratory and invasive properties, playing a crucial role in cancer progression, and it is considered a potential target for cancer therapy [48]. A genome-wide association may represent either a direct gene effect or an indirect effect through modifying genes influenced by environmental risk factors. Most EC and BC are estrogen-dependent tumors, and potential modifying gene effects could arise from variations in the estrogenic pathway. None of the genes in the novel EC–BC loci have been previously reported as hormonal modifiers, but the G protein-coupled estrogen receptor signaling has been suggested to suppress *PIEZO1*, which may influence proliferation [49]. Interestingly, *PIEZO1* and *WNT3* are often concurrently upregulated, and activation of *WNT3* has been implicated in both BC and EC [50,51,52]. These relationships suggest that estrogen may modulate gene regulation and activate pathways, such as *WNT3*, contributing to the progression of both cancers. Furthermore, the nearby *CTU2* gene has been implicated in hepatocellular carcinoma development [53]. Although the majority of EC and BC are hormone-dependent, there are triple negative BC and non-hormonal dependent HER2positive BC as well as type 2 EC which are not [54,55]. This study is based on unselected EC and BC cases; further studies on selected hormone-dependent EC and BC cases could be a future approach.

The locus at 17q11.2 has been highlighted in previous LOH studies related to BC [36]. Additionally, a study of Swedish high-risk BC families (≥3 first or second degree relatives of BC) identified a shared haplotype in this region [56], which was further supported by a sequence variant in the exome of the gene *ASIC2* (previously known as *ACCN1*) [56]. The *ASIC2* gene has been suggested to be involved in metastasis in triple-negative breast cancer and shown to promote metastasis in colorectal cancer through activation of the calcineurin/NFAT1 axis [57,58]. Another study highlighted a novel mechanism regulating IL-11 expression in endometrial adenocarcinoma cells via a prostaglandin receptor and calcineurin-1 [59]. Given that estrogen upregulates calcium-related proteins, this pathway could represent a potential link between BC and EC. The locus at 17q11.2 has previously been reported to be associated with EC and BC separately but at a distance from the current locus. An EC GWAS identified a susceptibility locus located 2 Mb away [12]. It is unlikely, but not excluded, that these represent the same genetic risk locus. Functional studies, such as mouse models, with gene knockout via CRISPR-Cas9 or modulating RNA expression with RNA interference, could be valuable methods for investigating the roles of these specific genes in cancer progression. Additionally, RNA expression analysis of tumor samples could also provide useful insights.

In our previous haplotype GWASs, we identified novel susceptibility loci with higher ORs, ranging from 1.27 to 3.6, than those generally reported in GWAS of single variants, a finding that is also supported by this study [18,20,60,61]. The higher ORs may be attributed to the multiallelic haplotype strategy, which enables the capture of the associated region of interest—specifically, an SNP with designated neighboring variants (a haplotype). Since haplotype analyses require homogeneous populations, this is not always feasible; however, it has been achieved with these Swedish samples. The significance level can be debated, given that multiple tests were conducted in each window. In this study, no corrections for multiple testing were performed, as the haplotypes of various lengths within a region represent the same genetic loci. This can be exemplified by the locus 8p21.2, where eight significant (along with several insignificant) haplotypes with similar odds ratios share a common central haplotype. A similar phenomenon is observed at all three loci, although the number of significant haplotypes varies. We propose that the haplotype with the lowest *p*-value most accurately delineates the area of interest (Table 1). The underlying biological mechanism is likely associated with the cycles of homologous recombination that occur, which subsequently “condense” the genetic risk. Therefore, conventional genome-wide significance, with a *p* < 5 × 10^−8^, was applied.

The strength of this study is the relatively homogenous and large Swedish EC and BC cohorts which enabled comprehensive haplotype analysis. In addition, all SNPs and haplotypes were examined without preconceived assumptions, highlighting potential areas that previously targeted GWASs or limited analyses of known BC regions may have missed. However, a limitation of this study is that it included individuals affected by either EC or BC, without considering other cancers or family history of cancer among the cases. Ideally, an equal number of EC and BC cases should have been selected for this syndrome approach GWAS. The lower number of EC cases (*n* = 1133) compared to BC cases (*n* = 3215) in this study may have introduced bias and led to an underestimation of loci associated with a higher risk for EC than for BC. Furthermore, a concern regarding the reproducibility of these findings in other populations is the known global diversity in genetic backgrounds [62]. This study is based on Swedish cases and controls. Therefore, future haplotype GWASs in diverse populations are warranted.

## 4. Materials and Methods

### 4.1. Study Population

In this study, we included 3215 invasive BC cases, 1133 invasive EC cases, and 5032 controls (Figure 2). The BC cases were obtained from three Swedish cohorts: KARMA (*n* = 2712), KARBAC1 (*n* = 394) and KARBAC2 (*n* = 109) [14,63,64]. The KARMA cohort is a Swedish population-based cohort that underwent a screening or clinical mammogram in Sweden between October 2010 and March 2013 [63]. KARBAC1 (N = 394) is a Swedish hospital-based cohort of consecutive breast patients recruited from October 1998 to May 2000 [64]. KARBAC2 (N = 109) is a cohort with germline BRCA1/BRCA2 negative Swedish BC cases recruited from a clinical genetic counselling department between February 2000 and January 2012 [14]. The EC cases were obtained from two Swedish cohorts—RENDOCAS (*n* = 555) and CAHRES (*n* = 578) [65,66]. RENDOCAS is a hospital-based cohort of Swedish consecutive EC cases who underwent surgery between 2008 and 2011 [65]. CAHRES is a Swedish population-based nationwide cohort recruited between January 1994 and December 1995 [66]. The controls were sourced from the KARMA cohort, consisting of healthy Swedish women aged 40–74 years who underwent mammographic screening. All five cohorts have been previously analyzed in single nucleotide GWAS for EC or BC separately as part of the global collaborations of BCAC and ECAC [11,12]. Based on our previous Swedish haplotype GWAS with comparable sample sizes of endometrial, breast, and colorectal cancers, we were confident that the study was adequately powered.

All studies were approved by the local ethical boards, and all individuals gave written informed consent. Specifically, the approvals were as follows: KARMA—Regional Ethics Committee, Stockholm, Dnr 2010/958-31/1; KARBAC1—Ethical Committee of Karolinska Institutet, Dnr 98-232; KARBAC2—Regional Ethical Committee, Stockholm, Dnr 2010/1156-31/2 Dnr 2012/1453-32 and 2011/1686-32; RENDOCAS—Regional Ethical Committee, Stockholm, Dnr 2010/1536-31/2; CAHRES—Ethical Committee of Karolinska Institutet, Dnr 98-036.

### 4.2. Genotyping and Quality Control

Individuals were genotyped using the Illumina Infinium OncoArray-500K B Bead-Chip. The five cohorts shared a total of 483,972 SNPs, and the datasets were merged using PLINK v1.9. During quality control (QC), 2332 variants were excluded due to a genotype call rate of less than 98% (Geno 0.02), 138,834 variants were removed for having a minor allele frequency of less than 0.01 (MAF 0.01), and 634 variants were excluded for deviating from the Hardy–Weinberg equilibrium at *p* < 0.001 (HWE 0.001). No individual was excluded due to missing genotype data (Figure 2).

To identify ethnic outliers, a multidimensional scaling (MDS) was performed on the remaining individuals and variants across four dimensions using predefined MDS coordinates 1 (C1), C2, C3 or C4. Samples with MDS values above +0.04 or below −0.04 were classified as ethnic outliers and subsequently excluded from the analysis (15 BC cases, 17 EC cases, and 11 controls). The final dataset consisted of 4316 cases (1116 EC and 3200 BC), 5021 controls, and a total of 332,906 variants (Figure 2). The reference genome panel GRCh37 was utilized.

### 4.3. Statistical Methods

To examine the associations between haplotypes of various lengths (exposure) and the outcomes of EC and BC, we conducted a sliding window haplotype GWAS with window sizes ranging from 1 to 25 SNPs in PLINK v1.07 [67]. The sliding window strategy was chosen to define the candidate region of interest from the first to the last SNP, with window sizes ranging from 1 to 25 SNPs. This range was chosen based on previous studies, which showed that the sizes of the candidate regions varied, making this range appropriate [18,19,20]. We applied the default setting of minor haplotype frequency threshold of 0.01 in PLINK v.1.07, meaning that each haplotype or SNP with a frequency above 1% was tested individually against all other SNPs/haplotypes in this window exceeding this threshold [67,68].

To adjust for population stratification, we used a logistic regression model, where the MDS coordinates C1–C4 (refer to “Genotyping and quality control”) were included as covariates. ORs and *p*-values were calculated using the default settings for haplotype analysis in PLINK v1.07 [67]. The 95% confidence intervals for the haplotypes reported in Table 1 were calculated manually from PLINK output based on OR and “STAT”. The aim of this study was to identify candidate loci associated with EC and BC risk, so only loci with OR > 1 were reported. A genome-wide significance level of *p* < 5 × 10^−8^ was considered statistically significant [69]. The definition of a shared EC–BC candidate risk locus is statistically significant in the present combined analysis, with a lower *p*-value compared to previous individual analyses of EC and BC. No further correction for multiple testing beyond the genome-wide significance level was performed, as we assumed that all haplotypes of various lengths within each sublocus represented the same genetic risk locus.

## 5. Conclusions

In conclusion, this combined haplotype GWAS of EC and BC samples identified three loci associated with a shared genetic risk, two of which are novel. Further studies are warranted to replicate these findings, identify the variants, and integrate them into a polygenic risk score for use in clinical practice.

## Figures and Tables

**Figure 1 ijms-26-05461-f001:**
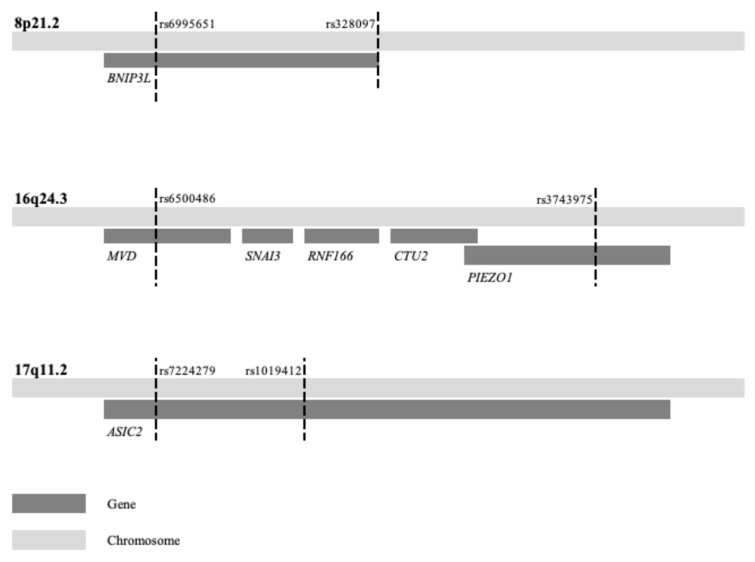
Three loci associated with combined endometrial and breast cancer risk. The first and last SNP of the haplotypes listed in Table 1 are here indicated by dashed lines and variant rs-name along with gene(s) within the locus.

**Figure 2 ijms-26-05461-f002:**
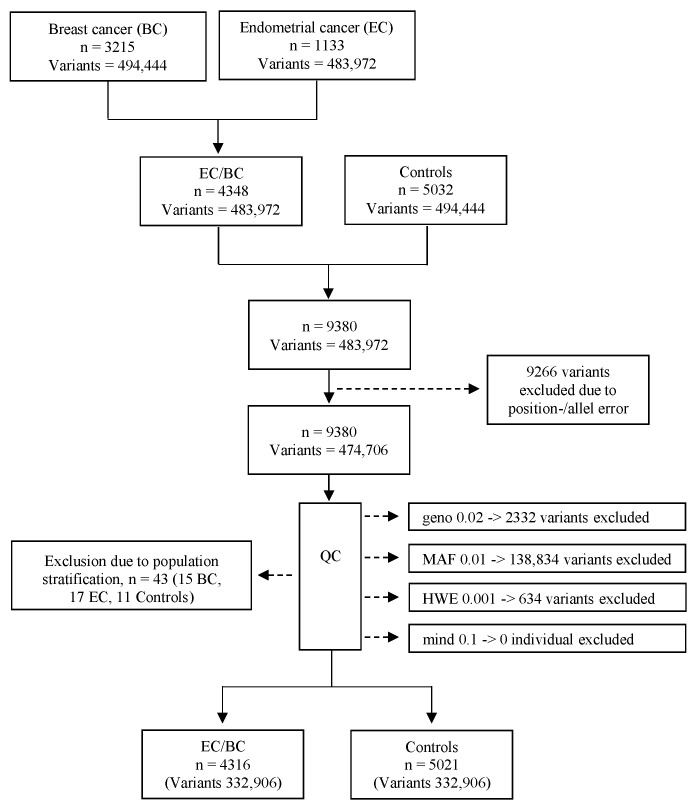
Flowchart of included individuals and variants. *n* = individuals; Geno 0.02 = excluded variants with a genotyping rate < 2%; MAF 0.01 = excluded variants with a minor allele frequency < 0.01; HWE 0.001 = excluded variants that failed the Hardy–Weinberg equilibrium test at *p* < 0.001; mind 0.1 = excluded individuals with missing genotypes > 10%.

**Table 1 ijms-26-05461-t001:** Statistically significant risk loci in a combined endometrial and breast cancer GWAS.

Locus	Gene(s)	SNP1–SNP2 (BP1–BP2)	Haplotype	F	OR (95% CI)	*p*-Value
8p21.2	*BNIP3L*	rs6995651–rs328097 (26300936–26343095)	AGGGGCGGAAGGA	0.02	2.1 (1.6–2.6)	1.6 × 10^−8^
10q26.13 *	*FGFR2*	rs45631627–rs2912778 (123338552–123338654)	GG	0.41	1.3 (1.2–1.4)	2.8 × 10^−15^
11q13.3 *	-	rs680618–rs614367 (69316881–69328764)	GGAAAAGGGA	0.11	1.4 (1.2–1.5)	9.7 × 10^−11^
16q12.1 *	*TOX3*	rs12918816–rs12929984 (52560213–52562811)	AG	0.26	1.3 (1.2–1.4)	4.2 × 10^−14^
16q24.3	*MVD*, *SNAI3*, *RNF166*, *CTU2*, *PIEZO1*	rs6500486–rs3743975 (88723629–88822297)	ACAGCCGGAAAAGGGGGAGGAGAA	0.01	2.4 (1.8–3.3)	3.8 × 10^−8^
17q11.2	*ASIC2*	rs7224279–rs1019412 (31450715–31488753)	GGAACGCG	0.11	1.3 (1.2–1.4)	4.3 × 10^−8^

Each genetic locus is presented with a gene in the area (if applicable), the first (SNP1) and last (SNP2) SNP, the first (BP1) and last (BP2) genomic positions, the haplotype with lowest *p*-value at each locus (window size 1–25), frequency (F), odds ratio (OR), and *p*-value. The reference genome panel used is GRCh37. * Previously published breast cancer locus [18].

**Table 2 ijms-26-05461-t002:** Results from combined and individual endometrial and breast cancer GWASs.

	ECBC GWAS (*n* = 4316)		BC GWAS (*n* = 3200)		EC GWAS (*n* = 1116)	
Locus	OR	*p*-Value	OR	*p*-Value	OR	*p*-Value
**8p21.2**	**2.01**	**1.6 × 10^−8^**	**2.08**	**3.9 × 10^−8^**	**1.95**	**3.8 × 10^−4^**
10q26.13	1.27	2.8 × 10^−15^	1.30	1.0 × 10^−20^	1.04	0.41
11q13.3	1.36	9.7 × 10^−11^	1.44	6.4 × 10^−13^	1.14	0.099
16q12.1	1.29	4.1 × 10^−14^	1.37	1.7 × 10^−18^	1.05	0.35
**16q24.3**	**2.40**	**3.9 × 10^−8^**	**2.27**	**1.5 × 10^−6^**	**2.84**	**1.8 × 10^−6^**
**17q11.2**	**1.27**	**4.3 × 10^−8^**	**1.26**	**1.4 × 10^−6^**	**1.32**	**4.0 × 10^−5^**

ECBC = results from the combined endometrial and breast cancer GWAS; BC = results from the previous individual breast cancer GWAS [18]; EC = results from the previous individual endometrial cancer GWAS [19]. The same EC and BC samples were included in both the combined and individual analyses. Loci highlighted in bold had lower *p*-values in the combined analysis compared to the individual analyses, suggesting a shared genetic risk for both EC and BC.

## Data Availability

The original data presented in the study are openly available in Zenodo at https://doi.org/10.5281/zenodo.15239435. Due to the Excel-sheet row limitation, the complete data are not included. The complete data of this study are available from the corresponding authors upon request.

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
