# Peer review of "Novel Shared Heritable Candidate Risk Loci of Breast and Endometrial Cancer—A Swedish Haplotype Genome-Wide Association Study"

_ijms, 2025, doi:10.3390/ijms26125461_

Round 1
Reviewer 1 Report
Comments and Suggestions for Authors
In the study entitled "Novel Shared Heritable Candidate Risk Loci of Breast and Endometrial Cancer, a Swedish Haplotype Genome-Wide Association Study" Barnekow and Lindblom have identified shared germline genetic risk loci for the breast and endometrial cancers that share both hormonal and environmental risk factors. The results from this merged analysis were compared with previous individual analyses of the same samples using the haplotype GWAS. This combined GWAS of endometrial and breast cancers identified three loci associated with shared genetic risk, out of which two were novel: 16q24.3 and 17q11.2.
The key comments are as below:
- The authors have clearly explained the methodology and the pipeline used in the analysis.
- The sample size is decent but there is a significant difference between the sample size of the EC versus the BC. Although, the authors have mentioned the limitations but the statistical bias in the sample size should also be mentioned in the limitations.
- In the line "Both EC and BC are estrogen-dependent tumours, and potential modifying gene effect could arise from variations in the estrogenic pathway" in the discussion, the authors must add the clarification in more detail. The authors must mention in details the different types of EC as the estrogen expression is significantly different between type 1 and type 2 EC PMID: 32932889.
- It would be worthy of discussing the major pathways and genes associated downstream of the three loci discovered by the authors. This can strengthen the association and links with other cancers.
Author Response
Thank you for your very thorough review of our paper and your valuable comments to improve the manuscript. Please see the attached file for responses to your comments.

Reviewer 2 Report
Comments and Suggestions for Authors
I've reviewed the article “Novel Shared Heritable Candidate Risk Loci of Breast and Endometrial Cancer, a Swedish Haplotype Genome-Wide Association Study” presented by Barnekow et al., well, The study employs a robust design, utilizing a combination of a haplotype GWAS approach to identify shared genetic risk loci for breast and endometrial cancer. However, given that this is a population-based study, it would be helpful to discuss how the sample sizes for both cancers were determined. Was a power analysis conducted to ensure that the study was adequately powered to detect the shared genetic loci between both cancers?
Major Comments:
The ms highlights three novel loci associated with breast and endometrial cancers. Although this is a significant finding, it would be valuable to see how these loci were validated. Were these loci replicated in an independent cohort or through another genomic validation method (e.g., qPCR, sequencing)?
The authors discusses using a sliding window haplotype analysis with the PLINK software. It would be beneficial to include more details about the parameters used for the analysis, such as window size choices, thresholds for significance, and the reasoning behind them. Moreover, were other statistical methods like the Bonferroni correction or false discovery rate (FDR) adjustment applied to account for multiple comparisons?
While the ms provides a solid analysis of the loci identified, it would be beneficial to elaborate more on the biological relevance of the loci. For instance, could the genes near these loci (such as BNIP3L, SNAI3, and PIEZO1) be involved in pathways regulating tumor progression, such as cell apoptosis or EMT? Expanding the discussion on how these loci may contribute to both breast and endometrial cancer risk at the molecular level could strengthen the manuscript.
It would be interesting discuss deeper validation studies to better understand how the identified genetic loci influence the development of breast and endometrial cancers. Could CRISPR-CAS9 or RNA interference experiments help investigate the role of specific genes (like BNIP3L or PIEZO1) in cancer progression?
A useful addition could be to analyze tumor samples from breast and endometrial cancer patients to see if the identified loci are differentially expressed or associated with specific tumor subtypes or stages. This could help identify potential clinical applications for the genetic loci identified.
Since both breast and endometrial cancers are estrogen-dependent, a discussion on how estrogen-related genetic variations or pathways might influence these shared risk loci would be helpful. Could variations in estrogen receptor signaling modify the effects of these loci?
The tables 1 and 2 could benefit from clearer headings and column descriptions. It might be helpful to indicate which genetic loci were previously identified and which are novel, perhaps through a separate column. Also, a forest plot graphic could visually confirms a strong positive associations between the genetic loci and cancer risk
Minor Comments:
The figure 1 does a good job of representing the loci associated with combined endometrial and breast cancer risk. However, it would be helpful to label the axes more clearly and include a more detailed legend.
The ms makes some strong claims regarding previous findings in this area but lacks some references when discussing specific genetic loci like BNIP3L, SNAI3, and PIEZO1. Adding citations for these genes and their connection to other type of cancer would provide further weight to the findings.
It would be helpful to briefly mention the potential clinical implications of these findings. For example, could the identified loci be developed into genetic tests for breast and endometrial cancer risk prediction?
Author Response
Response to Reviewer 2 Comments
Summary
I've reviewed the article “Novel Shared Heritable Candidate Risk Loci of Breast and Endometrial Cancer, a Swedish Haplotype Genome-Wide Association Study” presented by Barnekow et al., well, The study employs a robust design, utilizing a combination of a haplotype GWAS approach to identify shared genetic risk loci for breast and endometrial cancer.
Thank you very much for taking the time to review this manuscript. Please find the responses below and the corresponding corrections highlighted in “track changes” in the re-submitted manuscript file.
Comment 1: However, given that this is a population-based study, it would be helpful to discuss how the sample sizes for both cancers were determined. Was a power analysis conducted to ensure that the study was adequately powered to detect the shared genetic loci between both cancers?
Response 1: We selected all Swedish cases previously included in the single nucleotide GWASs from the collaboration of ECAC and BCAC. Based on our previous Swedish haplotype GWAS with comparable sample sizes of endometrial, breast, and colorectal cancers we were confident that the study was adequately powered to detect significant loci.
We added the text (in red) to Introduction page 2, line 61-63:
“By applying the same method and a comparable sample size, we conducted a haplotype GWAS to identify chromosomal regions that may harbor potential shared susceptibility loci for EC and BC.”
We also added text to Material and Methods, section 4.1 Study population, page 6, line 238-242:
“All five cohorts have been previously analyzed in single nucleotide GWAS for EC or BC separately as part of the global collaborations of BCAC and ECAC [11, 12]. Based on our previous Swedish haplotype GWAS with comparable sample sizes of endometrial, breast, and colorectal cancers we were confident that the study was adequately powered.”
Major Comments
Comment 2: The ms highlights three novel loci associated with breast and endometrial cancers. Although this is a significant finding, it would be valuable to see how these loci were validated. Were these loci replicated in an independent cohort or through another genomic validation method (e.g., qPCR, sequencing)?
Response 2: We agree that replication of the three novel candidate EC-BC loci, along with complementary analyses such as sequencing and functional studies, is warranted for future research. We added the text to Discussion page 4, line 112-114:
“Replication of the three novel candidate EC-BC loci, along with complementary analyses such as sequencing and functional studies, is warranted for future research.”
Comment 3: The authors discusses using a sliding window haplotype analysis with the PLINK software. It would be beneficial to include more details about the parameters used for the analysis, such as window size choices, thresholds for significance, and the reasoning behind them. Moreover, were other statistical methods like the Bonferroni correction or false discovery rate (FDR) adjustment applied to account for multiple comparisons?
Response 3: We agree, regarding the sliding window analysis and the window sizes we added the text to Material and Methods, section 4.3 Statistical Methods, page 7-8, line 274-277:
“The sliding window strategy was chosen to define the candidate region of interest from the first to the last SNP, with window sizes ranging from 1 to 25 SNPs. This range was chosen based on previous studies, which showed that the sizes of the candidate regions varied, making this range appropriate [18-20].”
Regarding the threshold for significance, there is a section in Discussion, page 5-6, line 196-206 and in Material and Methods, page 8, line 291-293 describing the background why we have not corrected for multiple testing beyond the genome-wide significance.
Comment 4: While the ms provides a solid analysis of the loci identified, it would be beneficial to elaborate more on the biological relevance of the loci. For instance, could the genes near these loci (such as BNIP3L, SNAI3, and PIEZO1) be involved in pathways regulating tumor progression, such as cell apoptosis or EMT? Expanding the discussion on how these loci may contribute to both breast and endometrial cancer risk at the molecular level could strengthen the manuscript.
Response 4: We agree. We added the text regarding locus 8p, page 4, line 138-143:
“The mitophagy receptor BNIP3 is highly expressed in various cancers, including BC and EC, and functions as a tumor suppressor gene [40, 41]. The gene BNIP3L (BNIP3-like) shares similar properties with BNIP3 [40, 42]. One study using HCT116 cells demonstrated that BNIP3 is activated the estrogen receptor beta (ERβ), promoting autophagy [43]. Furthermore, ERβ-mediated degradation of CyclinD1 was suggested to inhibit colon cancer cell growth via autophagy [43].”
Regarding the locus 16q, we added the text to Discussion, page 4-5, line 146-162:
“Among them, MVD, a member of the mevalonate pathway, interacts with RAC3 in HER2-positive breast cancer [46]. When hypomethylated, RAC3 can promote cell proliferation and invasion by increasing FASN expression in EC, potentially serving as a link between these two cancer types. [47]. The genes SNAI3 and PIEZO1, which are known to promote epithelial-mesenchymal transition (EMT), are noteworthy. EMT is a process that enables epithelial cells to acquire migratory and invasive properties, playing a crucial role in cancer progression and is considered a potential target for cancer therapy [48]. A genome-wide association may represent either a direct gene effect or an indirect effect through modifying genes influenced by environmental risk factor. Most EC and BC are estrogen-dependent tumors, and potential modifying gene effects could arise from variations in the estrogenic pathway. None of the genes in the novel EC-BC loci have been previously reported as hormonal modifiers but the G protein-coupled estrogen receptor signaling has been suggested to suppress PIEZO1, which may influence proliferation [49]. Interestingly, PIEZO1 and WNT3 are often concurrently upregulated, and activation of WNT3 has been implicated in both BC and EC [50-52]. These relationships suggest that estrogen may modulate gene regulation and activate pathways such as WNT3, contributing to the progression of both cancers.”
Regarding the locus 17q we added the text to Discussion, page 5, line 172-177:
“ASIC2 gene has been suggested to be involved in metastasis in triple-negative breast cancer and shown to promote metastasis in colorectal cancer through activation of the calcineurin/NFAT1 axis [57, 58] Another study highlighted a novel mechanism regulating IL-11 expression in endometrial adenocarcinoma cells via a prostaglandin receptor and calcineurin-1 [59]. Given that estrogen upregulates calcium-related proteins, this pathway could represent a potential link between BC and EC.”
Comment 5: It would be interesting discuss deeper validation studies to better understand how the identified genetic loci influence the development of breast and endometrial cancers. Could CRISPR-CAS9 or RNA interference experiments help investigate the role of specific genes (like BNIP3L or PIEZO1) in cancer progression? A useful addition could be to analyze tumor samples from breast and endometrial cancer patients to see if the identified loci are differentially expressed or associated with specific tumor subtypes or stages. This could help identify potential clinical applications for the genetic loci identified.
Response 5: Yes, we agree. We added the text to Discussion, page 5, line 180-184:
“Functional studies, such as mouse models with gene knockout via CRISPR-Cas9 or modulating RNA expression with RNA interference, could be valuable methods for investigating the roles of these specific genes in cancer progression. Additionally, RNA expression analysis of tumor samples could also provide useful insights.”
Comment 6: Since both breast and endometrial cancers are estrogen-dependent, a discussion on how estrogen-related genetic variations or pathways might influence these shared risk loci would be helpful. Could variations in estrogen receptor signaling modify the effects of these loci?
Response 6: We agree! We added the text and reference for clarification to the 8p section in Discussion, page 4, line 140-143:
“One study using HCT116 cells demonstrated that BNIP3 is activated by the estrogen receptor beta (Erβ), promoting autophagy [43]. Furthermore, ERβ-mediated degradation of CyclinD1 was suggested to inhibit colon cancer cell growth via autophagy [43].”
We also added the text and references to 16q section in Discussion page 5, line 156-162:
“None of the genes in the novel EC-BC loci have been previously reported as hormonal modifiers but the. G protein-coupled estrogen receptor signaling has been suggested to suppress PIEZO1, which may influence proliferation [49]. Interestingly, PIEZO1 and WNT3 are often concurrently upregulated, and activation of WNT3 has been implicated in both BC and EC [50-52]. These relationships suggest that estrogen may modulate gene regulation and activate pathways such as WNT3, contributing to the progression of both cancers.”
We also added the text and reference to the 17q section in Discussion, page 5, line 174-177:
“Another study highlighted a novel mechanism regulating IL-11 expression in endometrial adenocarcinoma cells via a prostaglandin receptor and calcineurin-1 [59]. Given that estrogen upregulates calcium-related proteins, this pathway could represent a potential link between BC and EC.”
Comment 7: The tables 1 and 2 could benefit from clearer headings and column descriptions. It might be helpful to indicate which genetic loci were previously identified and which are novel, perhaps through a separate column. Also, a forest plot graphic could visually confirms a strong positive associations between the genetic loci and cancer risk
Response 7: Thank you, we considered a forest plot but decided to add an asterisk in Table 1 for the previously published loci on chromosomes 10, 11, and 16. In Table 2, the loci in bold are the novel EC-BC loci.
Minor Comments:
Comment 8: The figure 1 does a good job of representing the loci associated with combined endometrial and breast cancer risk. However, it would be helpful to label the axes more clearly and include a more detailed legend.
Response 8: Thank you. We included a more detailed legend and an illustration of the light yellow “Chromosome” below the Gene illustration. The added text in Figure 1 legend (in red), page 3, line 97-99:
“Figure 1. Three loci associated with combined endometrial and breast cancer risk. The first and last SNP of the haplotypes in Table 1 are indicated by dashed lines and variant rs-name along with gene(s) within the locus.”
Comment 9: The ms makes some strong claims regarding previous findings in this area but lacks some references when discussing specific genetic loci like BNIP3L, SNAI3, and PIEZO1. Adding citations for these genes and their connection to other type of cancer would provide further weight to the findings.
Response 9: We added references for BNIP3L, SNAI3, and PIEZO1, please see response 4.
Comment 10: It would be helpful to briefly mention the potential clinical implications of these findings. For example, could the identified loci be developed into genetic tests for breast and endometrial cancer risk prediction?
Response 10: Thanks, we added the text (in red) to Conclusion, page 8, line 296-299.
“Further studies are warranted to replicate these findings, identify the variants, and integrate them into a polygenic risk score for use in clinical practice.”
Reviewer 3 Report
Comments and Suggestions for Authors
Very well-written manuscript, with robust and concrete analyses. I only have 3 observations:
Study Population: Please detail the type of population included in the cohort or populations included. Are they young patients or those with certain criteria for hereditary cancers? Or are they cancers that are presumed to be non-hereditary? Or do they have no selection criteria, except for having cancer? This makes it difficult to determine which population the results can be inferred from.
In summary and in the conclusions, you mention "Further studies are warranted to replicate these findings and to explore the implications for cancer prevention." In this sense, it is not clear how haplotypes identified as risk factors in a population study can potentially be used in preventive strategies. This is also not mentioned in the discussions. I believe it can be mentioned that further analysis should be carried out to determine its pathophysiological or clinical implications, but I do not believe that "exploring the implications for cancer prevention" should not yet be mentioned, as this could generate a false perception of its usefulness and potential.
In discussions, mention the population of origin, and emphasize that studies in other populations may have different results, and that other populations of different ethnic origins should be studied.
Author Response
Response to Reviewer 3 Comments
Comment 1: Study Population: Please detail the type of population included in the cohort or populations included. Are they young patients or those with certain criteria for hereditary cancers? Or are they cancers that are presumed to be non-hereditary? Or do they have no selection criteria, except for having cancer? This makes it difficult to determine which population the results can be inferred from.
Response 1: Thank you for pointing this out. It is a mixed ECBC cohort from five cohorts. We agree that details of the cohorts should be included. We added the text (in red) to section “4.1 Study population”, page 6, line 224-240.
“The KARMA cohort is a Swedish population-based cohort who underwent screening or clinical mammogram in Sweden between October 2010 and March 2013 [53]. KARBAC1 (N = 394) is a Swedish hospital-based cohort of consecutive breast patients recruited from October 1998 to May 2000 [54]. KARBAC2 (N = 109) is a cohort with germline BRCA1/BRCA2 negative Swedish BC cases recruited from a clinical genetic counselling department between February 2000 and January 2012 [16]. The EC cases were obtained from two Swedish cohorts - RENDOCAS (n = 555) and CAHRES (n = 578) [55, 56]. RENDOCAS is a hospital-based cohort of Swedish consecutive EC cases who underwent surgery between 2008 and 2011 [55]. CAHRES is a Swedish population-based nationwide cohort recruited between January 1994 and December 1995 [56]. The controls were sourced from the KARMA cohort, consisting of healthy Swedish women aged 40–74 years who underwent mammographic screening. All five cohorts have been previously analyzed in single nucleotide GWAS for EC or BC separately as part of the global collaborations of BCAC and ECAC [12, 13].”
Comment 2: In summary and in the conclusions, you mention "Further studies are conclusion to replicate these findings and to explore the implications for cancer prevention." In this sense, it is not clear how haplotypes identified as risk factors in a population study can potentially be used in preventive strategies. This is also not mentioned in the discussions. I believe it can be mentioned that further analysis should be carried out to determine its pathophysiological or clinical implications, but I do not believe that "exploring the implications for cancer prevention" should not yet be mentioned, as this could generate a false perception of its usefulness and potential.
Response 2: Thanks, we agree that this ECBC finding warrants further investigation. We modified the text (in red) in the Abstract, page 1, line 30-32.
“Further studies are warranted to replicate these findings to determine its pathophysiological role and future clinical implications.”
….. and as well in the Conclusion, page 8, line 296-299.
“Further studies are warranted to replicate these findings, identify the variants, and integrate them into a polygenic risk score for use in clinical practice.”
Comment 3: In discussions, mention the population of origin, and emphasize that studies in other populations may have different results, and that other populations of different ethnic origins should be studied.
Response 3: Thanks, we agree that genetic diversity among ethnic populations exists, which can affect the external validity. We added the text (in red) to Discussion, page 6, line 216-219.
“Furthermore, a concern regarding the reproducibility of these findings in other populations is the known global diversity in genetic backgrounds [52]. This study is based on Swedish cases and controls. Therefore, future haplotype GWASs in diverse populations are warranted.”
Round 2
Reviewer 2 Report
Comments and Suggestions for Authors
The authors have made changes to their manuscript and have addressed the questions.